# Mapping the distribution of sandflies and sandfly-associated pathogens in China

**Xue-Geng Hong**[1☯], **Ying Zhu**[2☯], **Tao Wang**[3], **Jin-Jin Chen**[1], **Fang Tang**[4], **Rui-Ruo Jiang**[5], **Xiao-Fang Ma**[6], **Qiang Xu**[1], **Hao Li**[1,2], **Li-Ping Wang**[7], **Yi Sun**[1]\*, **Li-Qun Fang**[1]\*, **Wei Liu**[1]\*

**1** State Key Laboratory of Pathogen and Biosecurity, Academy of Military Medical Science, Beijing, P. R. China, **2** Department of Epidemiology and Biostatistics, School of Public Health, Wuhan University, Wuhan, P. R. China, **3** The 949th Chinese PLA Hospital, Altay, P. R. China, **4** Center for Disease Control and Prevention of Chinese People's Armed Police Forces, Beijing, P. R. China, **5** Institute of NBC Defense, PLA Army, Beijing, P. R. China, **6** Qingdao Municipal Center for Disease Control and Prevention, Qingdao, P. R. China, **7** Center for Public Health Surveillance and Information Service, Chinese Center for Disease Control and Prevention, Beijing, P. R. China

☯ These authors contributed equally to this work.

\* sunyi7310@sina.com (YS); fang_lq@163.com (L-QF); lwbime@163.com (WL)

## Abstract

### Background

Understanding and mapping the distribution of sandflies and sandfly-associated pathogens (SAPs) is crucial for guiding the surveillance and control effort. However, their distribution and the related risk burden in China remain poorly understood.

### Methods

We mapped the distribution of sandflies and SAPs using literature data from 1940 to 2022. We also mapped the human visceral leishmaniasis (VL) cases using surveillance data from 2014 to 2018. The ecological drivers of 12 main sandfly species and VL were identified by applying machine learning, and their distribution and risk were predicted in three time periods (2021–2040, 2041–2060, and 2061–2080) under three scenarios of climate and socio-economic changes.

### Results

In the mainland of China, a total of 47 sandfly species have been reported, with the main 12 species classified into three clusters according to their ecological niches. Additionally, 6 SAPs have been identified, which include two protozoa, two bacteria, and two viruses. The incidence risk of different VL subtypes was closely associated with the distribution risk of specific vectors. The model predictions also revealed a substantial underestimation of the current sandfly distribution and VL risk. The predicted areas affected by the 12 major species of sandflies and the high-risk areas for VL were found to be 37.9–1121.0% and 136.6% larger, respectively, than the observed range in the areas. The future global changes were projected to decrease the risk of mountain-type zoonotic VL (MT-ZVL), but anthroponotic VL (AVL) and desert-type zoonotic VL (DT-ZVL) could remain stable or slightly increase.

**Data Availability Statement:** The datasets supporting the conclusions of this article are included within the article and its appendices.

**Funding:** This work was supported by the National Science Foundation for Distinguished Young Scholars of China (81825019 to WL) and National Key Research and Development Program of China (2021YFC2302004 to LQF). The funders had no role in study design, data collection and analysis, decision to publish, or preparation of the manuscript.

**Competing interests:** The authors have declared that no competing interests exist.

## Conclusions

Current field observations underestimate the spatial distributions of main sandfly species and VL in China. More active surveillance and field investigations are needed where high risks are predicted, especially in areas where the future risk of VL is projected to remain high or increase.

### Author summary

Our research provides a comprehensive understanding of the distribution of sandflies and SAPs in China. We have identified 47 sandfly species and 6 SAPs, with 12 main sandfly species forming three distinct ecological clusters. Our machine learning models predict the distribution and risk of sandfly species and VL under various climate and socioeconomic scenarios. The model predictions reveal a significant underestimation of the current sandfly distribution and VL risk. Future global changes are projected to decrease the risk of MT-ZVL, but AVL and DT-ZVL could remain stable or slightly increase. These findings underscore the need for more active surveillance and field investigations of vectors, especially where the future risk of VL is projected to remain high or increase. This study provides important theoretical support for guiding the surveillance and control of sandflies and SAPs.

## 1. Introduction

Vector-borne diseases (VBDs) are estimated to account for over 17% of all known infectious diseases, resulting in at least 700,000 deaths annually [1]. Most of these diseases are classified as neglected tropical diseases (NTDs) due to their disproportionate impact on individuals in resource-poor countries within the tropics and subtropics [2]. Despite the focus of VBD research on arthropods like mosquitoes and ticks, other vectors such as sandflies and mites have been relatively overlooked [3,4]. Sandflies are hematophagous ectoparasites that are capable of infesting a wide range of mammals, including domestic animals, wildlife, and humans over a global range [5]. Known primarily as vectors for *Leishmania*, sandflies also transmit other sandfly-associated pathogens (SAPs), including bacteria and viruses. One example is Carrion's disease, a two-stage illness spread by sandflies and caused by the *Bartonella bacilliformis* bacteria, prevalent in Central and South America [6]. In addition to bacteria, sandflies also transmit certain arboviruses [7]. Among these, the Toscana virus (TOSV) stands out due to its propensity to infect the central and peripheral nervous system, potentially causing severe conditions such as meningitis or encephalitis [8]. Despite their impact, the public health consequences of these SAPs on populations in the endemic areas remain rarely investigated, largely due to limited research interest and lack of surveillance program.

As a country with environmental richness and megadiversity, China hosts environmental and sociodemographic aspects that render a high susceptibility to the establishment of sandfly population, and a complex composition of SAPs. However, due to a lack of research interest and monitoring programs, almost all research on SAPs has been focused on *Leishmania*, with very limited research on other SAPs. Among all the sandfly-borne diseases in China, visceral leishmaniasis (VL) remains as the deadliest one with significant disease burden, morbidity, and mortality [9]. According to the epidemiological features, three distinct types can be

defined: anthroponotic VL (AVL), mountain-type zoonotic VL (MT-ZVL), and desert-type ZVL (DT-ZVL) [10], which have heterogenic pattern in their distributions and epidemics. Historically, VL has imposed a significant disease burden in China. Although VL has maintained a low-level epidemic trend in recent years, showing a general downward trend, it's noteworthy that from 2015 to 2019, there was a resurgence of MT-ZVL in 13 historically-endemic counties, and the epidemic range has continuously expanded with sporadic outbreaks in some endemic areas [11].

Here we perform a comprehensive literature review on the occurrence of sandfly species and SAPs reported in China between 1940 and 2022. Additionally, we collected data on human cases of VL reported to the Chinese Information System for Disease Control and Prevention (CISDCP) between 2014 and 2018. Species distribution prediction models for the main sandfly species were constructed using machine learning methods. The predicted results of the vector sandflies were incorporated as predictors in the VL risk model, resulting in the predicted incidence of VL. We also predicted the vector distribution and VL risk in three scenarios integrating climate and socioeconomic changes, 2021–2040, 2041–2060 and 2061–2080. These scenarios followed two widely used global change frameworks: the Representative Concentration Pathways (RCPs) for climate change and the Shared Socioeconomic Pathways (SSPs) for socioeconomic change [12,13]. The three combined scenarios were SSP1-RCP2.6 (SSP126), SSP2-RCP4.5 (SSP245), and SSP5-RCP8.5 (SSP585).

## 2. Material and methods

### 2.1 Data collection and management

We assembled an occurrence database of sandflies and SAPs, comprising point location (i.e., associated with a specific latitude and longitude) or polygon (e.g., county or province) locations of their confirmed presence in the mainland of China between January 1940 and December 2022. Multiple sources were used: (1) a systematic search of PubMed, Web of Science, China WanFang database, and China National Knowledge Infrastructure databases for published literature in the English and Chinese language from January 1940 to December 2022. The search was structured to include data on the distribution of sandflies and detection of SAPs by using the terms "sandfly" or "sandflies" or "sand fly" or "sand flies" and "China"; (2) the recorded SAPs extracted from GenBank; (3) *Chinese Sandflies* composed by National Institute of Parasitic Diseases, Chinese Center for Disease Control and Prevention, which documents the distribution of sandflies and SAPs [14]; and (4) survey data on the locations of sandflies or SAPs from entomological surveys conducted by our group (Fig 1, Table A in S1 Appendix and S1 Data). We excluded studies that were: (1) letters to the editor, opinion and editorial articles, media reports, and abstracts of posters; (2) data of experimental studies without field investigations; (3) drug or vaccine trials for SAPs; (4) detection methods not specified or unclearly described for SAPs or (5) reporting ambiguous data on the geographic information (Fig 1 and Table B in S1 Appendix). Moreover, we obtained individual data on all laboratory-confirmed and clinically-confirmed VL cases from 2014 to 2018 from the CISDCP. Based on previous literature and scientific hypotheses, we collected explanatory variable data related to sandfly vectors and VL (Tables C and D in S1 Appendix) [5,10,15–17]. These variables included elevation (average and standard deviation), population density, rural population proportion, and rural population density, as well as bioclimatic variables (BIO 1–19). Following a correlation analysis of the 19 bioclimatic variables, we narrowed them down to a select group of eight (BIO 1–5, 12, 15 and 17) (Tables E in S1 Appendix). These chosen variables, in conjunction with the others, were then integrated into the final construction of our model. Details on data acquisition, screening, and processing are provided in Text A in S1 Appendix.

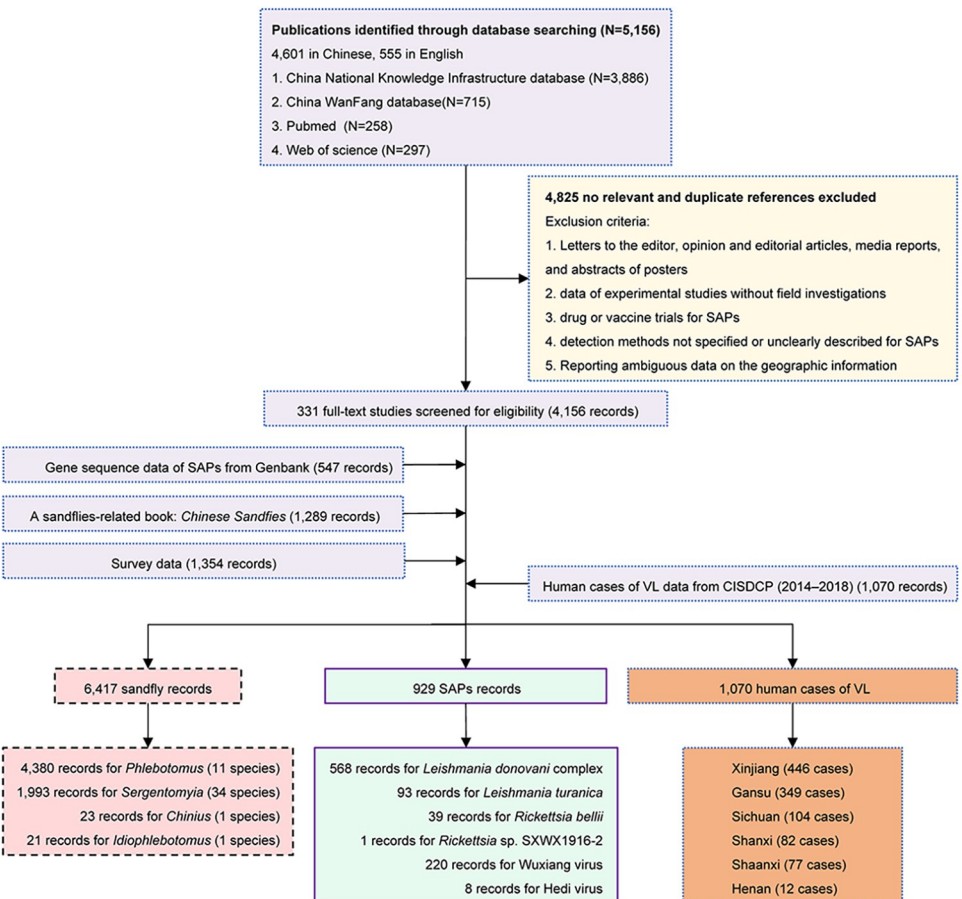

**Fig 1. The flow diagram of literature review.** SAPs sandfly-associated pathogens; VL visceral leishmaniasis; CISDCP Chinese Information System for Disease Control and Prevention.

## 2.2 Ecological modeling and clustering analysis of sandflies

Considering the large time span of the collected data and the significant difference of the ecological niche across the years, we only used sandfly distribution records that had been reported since 1980 for the modeling. Totally 12 dominant sandfly species that were found in more than 20 counties in China were used for boosted regression tree (BRT) modelling analysis. An occurrence was defined as one or more reported presence of the sandfly at a unique location (the same county). All surveyed counties were classified based on the presence or absence of sandflies. Specifically, counties with at least one recorded occurrence were marked as "presence", while those without any evidence of occurrence, despite being surveyed, were marked as "absence". Counties that had not been surveyed were excluded from model construction. For each BRT model, we set a threshold value based on the training data that maximized the sum of sensitivity and specificity along the receiver operating characteristic curve (ROC), and we took the average of this threshold value across 100 models [18,19]. Based on this threshold value, we categorized all counties into high-risk (fitted probability ≥threshold value) and low-risk (fitted probability <threshold value) groups. Details on the ecological modeling and control of sampling bias of sandflies are provided in Text B in S1 Appendix.

The similarities of ecological niches among 12 predominant sandfly species were assessed by performing a hierarchical clustering analysis using a weighted-average linkage methodology

[20]. The clustering patterns among the 12 sandfly species were shown by a dendrogram and a thematic matrix. We generated three indicators for the clustering of sandfly species. We defined the presence of a cluster as the presence of any sandfly species within that cluster, which were mapped at the county level. For details on the feature selection and generation for clustering, see Text B in S1 Appendix.

## 2.3 Ecological modeling of VL

To explore potential drivers of VL presence and incidence, we constructed a county-level two-stage extreme gradient boosting (XGBoost) model with a yearly temporal resolution in the R software package "xgboost" [21]. The modelling efforts were performed for two categories of VL based on their geographical distribution characteristics: AVL &DT-ZVL, which are prevalent in northwest China; MT-ZVL, which is concentrated in central China. At the first stage, a logistic XGBoost model was employed to fit the presence or absence of VL cases in each county. The presence or absence of VL was determined by comparing the model prediction values with the threshold value, which was chosen by maximizing the model's specificity and sensitivity, in accordance with the previous threshold selection method of sandfly model. At the second stage, the case incidence in the counties that had reported human cases from 2014 to 2018 was fitted with the gamma XGBoost model. The model-fitted incidence was calculated as follows: if the probability of presence predicted by the stage-1 logistic model was lower than the threshold value, then the predicted incidence was set to 0. Otherwise, the predicted incidence was set to the mean incidence predicted by the stage-2 gamma model.

To assess the goodness-of-fit of the two-stage XGBoost model, we calculated the observed average incidence of VL from 2014 to 2018. Based on the annual incidences, we first categorized the counties into three groups (annual incidence $<0.071/10^5$, no-risk, with outcome designated as 0; low-medium, $0.071$–$1.490/10^5$, designated as 1; and $\geq1.490/10^5$, high-risk designated as 2), where $0.071/10^5$ and $1.490/10^5$ represents the 25th and the 75th percentile of the county-level annual incidences from 2014 to 2018, respectively. We then compared it with the predicted VL incidence for the same period, 2014–2018. This comparison was presented as a misclassification matrix in which the fitted and observed categories were cross-tabulated (Table F in S1 Appendix). Details on the ecological modeling of VL are provided in Text C in S1 Appendix.

## 2.4 Predicting VL dynamics under the SSPs

Three scenarios were employed in this study, (1) SSP126, a low-emission scenario with a sustainable pathway; (2) SSP245, a medium-emission scenario with no significant deviation from historical trends; and (3) SSP585, a high-emission scenario with a fossil-fuel intensive pathway [12,13]. We processed variables under three scenarios in the periods 2021–2040, 2041–2060, and 2061–2080, obtaining variables for different scenarios and periods. These variables were then separately input into the previously constructed sandfly models to predict environmental suitability for different scenarios and periods. Based on fitted probabilities and threshold, counties were classified into presence and absence groups. We then calculated the changes in vector risk and incorporated the presence or absence of sandfly vectors as a future variable in the prediction of VL risk, along with other future variables. Lastly, we substituted the predicted vector variable and all other future variables into the previously constructed VL model to predict the incidence and changes in risk level of VL under different scenarios in the periods 2021–2040, 2041–2060, and 2061–2080. The grouping of the predicted risk level is consistent with the quartiles defined by the actual annual incidence rate previously.

## 3. Results

A total of 5,156 publications (555 in English and 4,601 in Chinese) were screened and 331 papers met the pre-set inclusion criteria, from which 4,156 records on sandflies and SAPs were extracted. Additional 3,190 records were extracted from GenBank (547), a sandflies-related book titled *Chinese Sandflies* (1,289), and a survey program performed by our group in the 1970s (1,354). We also obtained individual data on 1,070 laboratory-confirmed and clinically diagnosed VL cases from 2014 to 2018 from CISDCP. All these data taken together had composed a database of 8,416 records that are related to sandflies (6,417), SAPs (929), and human cases (1,070), respectively (Fig 1).

### 3.1 Distribution of sandfly species in the mainland of China

We compiled a database containing 6,417 unique records on geographical distribution of 47 known sandfly species in four genera that have been recorded (Fig 1 and S1 Data). The records are derived from 673 counties (24% of all counties in the mainland of China), which are mainly located within mid-to-high latitude (Figs A and B in S1 Appendix). The most widely distributed sandfly was *Phlebotomus* genus (recorded in 546 counties), followed by *Sergentomyia* genus (in 305 counties), *Idiophlebotomus* genus (in 8 counties), and *Chinius* genus (in 5 counties) (Figs C–E in S1 Appendix and S1 Data). At the species level, there are 12 predominant sandflies with their presence recorded in ≥20 counties. Among them the *Phlebotomus* (*P.*) *chinensis* and *P. mongolensis* belonging to *Phlebotomus* genus; *Sergentomyia* (*S.*) *squamirostris* belonging to *Sergentomyia* genus were the most widely distributed, each reported in over 100 counties, followed by *S. barraudi*, *P. kiangsuensis*, and *S. khawi*, each found in 50–100 counties (S1 Data). Sandfly species richness varied widely among seven biogeographic zones defined by climatic and ecological characteristics (Fig F in S1 Appendix). Central China and South China harbored the highest variety of sandfly species (28 and 19 species, respectively), in contrast to only three species observed in Northeast China. Forty prefectures (thirteen in Central China, twelve in Inner Mongolia-Xinjiang, nine in North China, four in South China, and two in Southwest China) reported over five species of sandfly, regarded as having high species richness (Fig F in S1 Appendix and S1 Data).

### 3.2 Environmental suitability and ecological clustering of major sandfly species

Ecological modeling for the 12 major sandfly species showed good prediction performance, with the average area under the curve (AUC) ranging from 0.916 to 0.996 (Table 1) and the testing partial AUC ratio ranging from 1.32 to 1.94 (Tables G and H in S1 Appendix). All the 12 major sandfly species were predicted to be present in a remarkably wider areas than was actually reported, with 72.4–1435.1% greater in the number of affected counties and 37.9–1121.0% wider in the affected geographical area, where 42.9–1228.3% larger population size is at potential risk (Table 1).

Among the four main vectors for VL, *P. chinensis* was the most widely distributed (found in 373 counties), also predicted as the most widely distributed, potentially affecting 435.9 million people in 893 counties. Three other species, *P. wui*, *P. longiductus*, and *P. alexandri*, were predicted to potentially affect 16.9, 15.8 and 15.0 million people in 69, 57 and 57 counties, respectively, significantly more than those actual records in 40, 33 and 25 counties, respectively (Table 1 and S1 Data).

Based on the significant ecological predictors obtained from the models, we grouped the 12 major sandfly species into three clusters with obvious spatial aggregation that showed distinct

**Table 1. The average AUC of the BRT models and predicted numbers, land areas and population sizes of affected counties for the 12 most prevalent sandfly species in the mainland of China.**

| Sandfly species | Average AUC (2.5–97.5% percentiles) | Predicted/observed (relative difference %) | | |
|---|---|---|---|---|
| | | Number of counties | Area (10,000 km²) | Population size (million) |
| P. chinensis[a,b,c,d] | 0.947 (0.941, 0.957) | 893/373 (139.4) | 144.9/77.4 (87.2) | 435.9/224.5 (94.1) |
| P. mongolensis | 0.968 (0.960, 0.976) | 588/239 (146.0) | 143.7/89.7 (60.1) | 280.6/150.7 (86.2) |
| P. kiangsuensis | 0.975 (0.960, 0.988) | 654/53 (1134.0) | 121.0/13.6 (789.9) | 371.3/41.3 (799.4) |
| P. wui[d] | 0.996 (0.993, 0.999) | 69/40 (72.5) | 144.3/102.3 (41.1) | 16.9/10.4 (62.7) |
| P. longiductus[d] | 0.996 (0.993, 0.999) | 57/33 (72.7) | 70.0/42.9 (63.2) | 15.8/9.8 (60.8) |
| P. alexandri[d] | 0.990 (0.983, 0.997) | 57/25 (128.0) | 141.8/39.3 (261.1) | 15.0/6.7 (124.0) |
| S. squamirostris[a,b,c] | 0.916 (0.885, 0.942) | 925/148 (525.0) | 158.2/28.9 (446.5) | 538.6/103.7 (419.3) |
| S. barraudi[a,b,c] | 0.954 (0.943, 0.961) | 875/57 (1435.1) | 158.4/13.0 (1121.0) | 463.1/34.9 (1228.3) |
| S. khawi | 0.963 (0.924, 0.980) | 174/45 (286.7) | 28.0/8.0 (250.5) | 80.3/25.6 (213.8) |
| S. koloshanensis | 0.982 (0.959, 0.991) | 321/32 (903.1) | 92.0/8.1 (1032.8) | 121.6/11.8 (927.9) |
| S. sinkiangensis | 0.995 (0.993, 0.998) | 50/29 (72.4) | 112.4/81.6 (37.9) | 12.0/8.4 (42.9) |
| S. bailyi | 0.994 (0.990, 0.996) | 101/26 (288.5) | 25.9/5.3 (389.1) | 37.1/10.5 (254.5) |

[a]Top 3 sandfly species potentially affecting the largest numbers of counties.

[b]Top 3 sandfly species potentially affecting the widest areas.

[c]Top 3 sandfly species potentially affecting the largest population sizes.

[d]Four sandfly vectors for VL.

The predicted numbers are compared with the actual observations from field surveys and the relative differences (%) are given in parentheses. AUC, area under the curve; BRT boosted regression tree; VL visceral leishmaniasis.

ecological features (Fig 2A–2D and Fig I in S1 Appendix). Cluster 1, which comprised *S. squamirostris*, *P. mongolensis*, *S. khawi* and *P. chinensis*, was predicted in 2,580 counties that cover biogeographic zones I–VI and almost all of zone II and large parts of zones III, V, and VI (Fig 2B). The Cluster 1 was ecologically characterized by moderate average annual temperatures and total precipitation and high temperature seasonality (Fig 2A). Cluster 2, which encompass *S. sinkiangensis*, *P. wui*, *P. alexandri* and *P. longiductus*, was predicted in the western part of biogeographic zone III and ecologically characterized by low total precipitation, large standard deviation of elevation, and significant mean diurnal range (Fig 2A and 2C). Cluster 3, which includes *S. bailyi*, *S. koloshanensis*, *P. kiangsuensis* and *S. barraudi*, was predicted to be mainly distributed in biogeographic V–VII, while being less predominant in zones II (Fig 2D). This cluster is ecologically characterized by low temperature seasonality, high mean annual temperature, and low mean diurnal range (Fig 2A).

### 3.3 Geographic distribution of SAPs

We compiled a database encompassing 929 records of six SAPs that belong to two *Leishmania* spp., i.e., *Leishmania* (*L.*) *donovani* complex and *L. turanica*; two *Rickettsia* spp., i.e., *Rickettsia* (*R.*) *bellii* and one uncharacterized *Rickettsia* SXWX1916-2; two *Phleboviruses*, i.e., Wuxiang virus (WUXV) and Hedi virus (HEDV). These SAPs were reported from six sandfly species in 29 counties, with *P. chinensis* harboring the highest number of SAPs, including one *Leishmania* species, two *Rickettsia* species, and two *Phleboviruses*, while the other five sandfly species were only associated with one or two *Leishmania* species (Figs 2E and 3A). In addition to sandflies, 14 animal species harboring four SAPs were determined in 66 counties (Figs 2E and 3B).

The causative pathogen of VL in China, *L. donovani* complex, which mainly consists of *L. donovani* and *L. infantum* [22], was determined to be the most widespread SAP, detected from

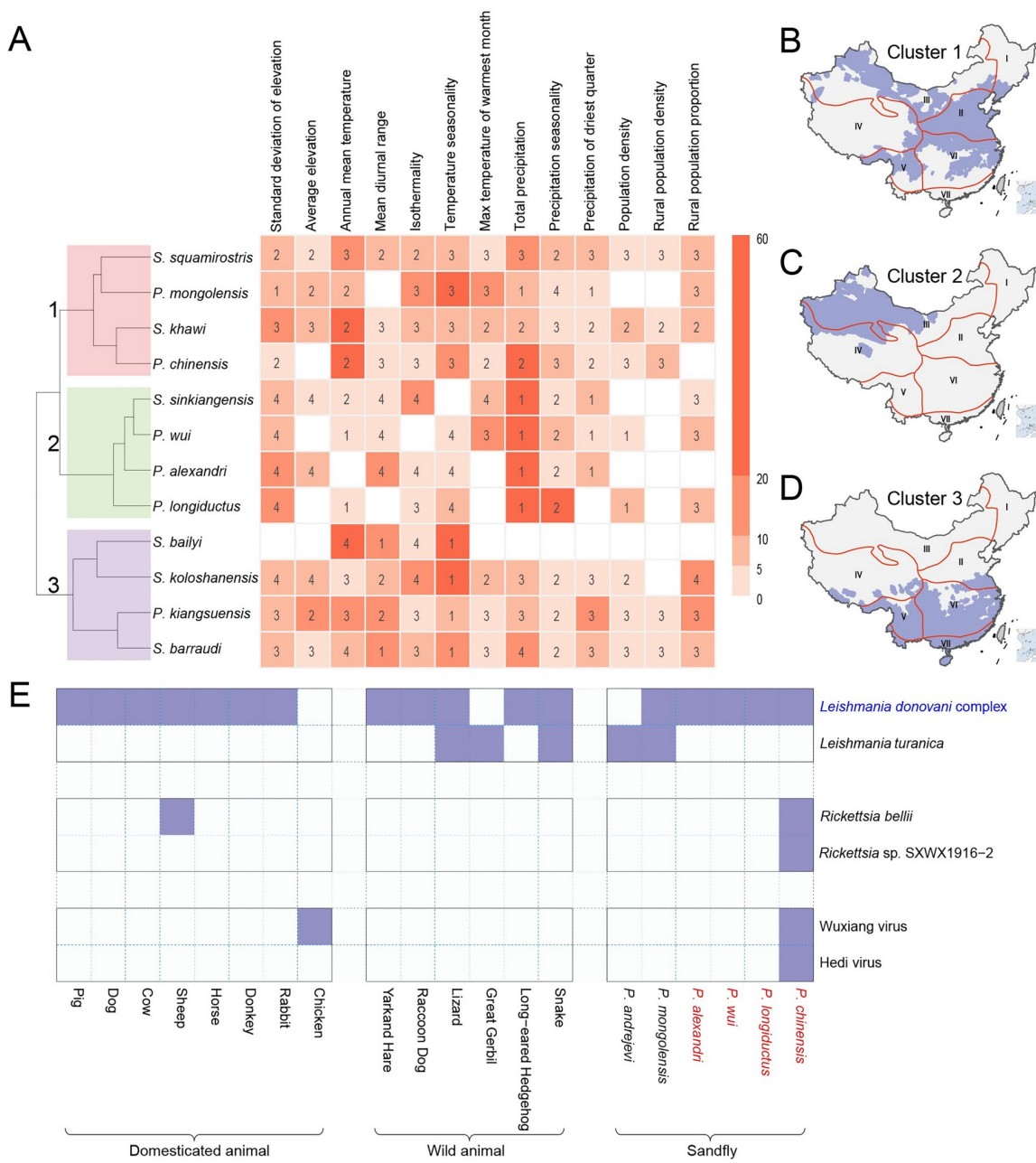

**Fig 2. Clustering of sandfly species, and the detection of SAPs in animals and sandflies. Panel A**, the dendrogram displays clusters 1–3 of sandfly species. The features used for clustering are the three properties associated with each predictor in the BRT model. Two of the three properties were shown to indicate possible levels of ecological suitability: i) the relative contributions of the predictors were shown in colors, with dark color corresponding to high contribution; ii) the standardized median value of the predictor was indicated by the numbers in the heatmap (numbers 1–4 indicate the position of this median in reference to the quartiles of this predictor among all counties). For details on the feature selection and generation for clustering, see Text B in S1 Appendix. **Panels B–D**, the spatial distribution of the three clusters of sandflies. Red solid lines indicate the boundaries of the seven biogeographic regions. **Panel E,** sandfly species and animals that carry SAPs. The names of the pathogens of VL are shown in blue, and the names of the vectors of VL are shown in red. SAPs sandfly-associated pathogens; BRT boosted regression tree; VL visceral leishmaniasis. Base layers of the maps were downloaded from Resource and Environment Science and Data Center (https://www.resdc.cn/DOI/DOI.aspx?DOIID=120).

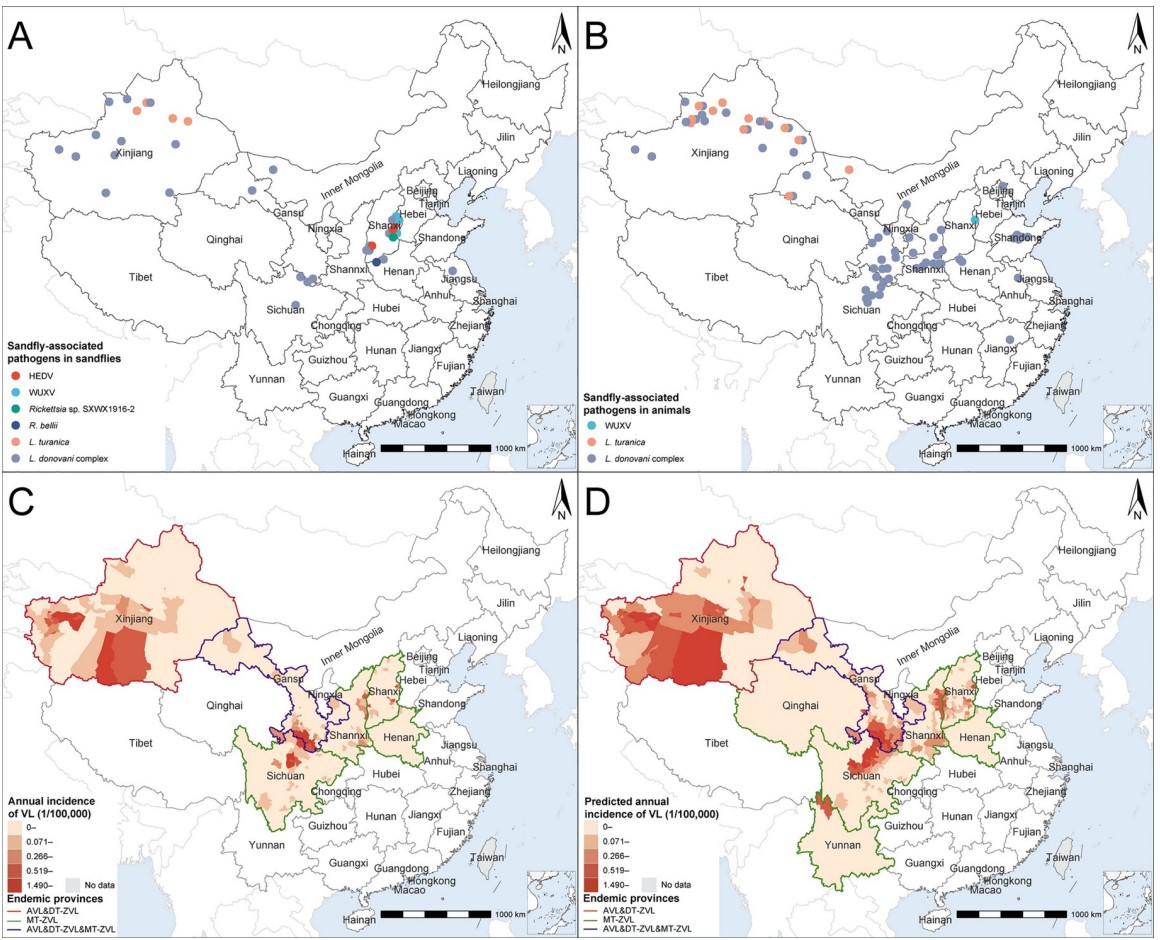

**Fig 3. The spatial distribution of SAPs, and the recorded and model-predicted annual incidence of VL at the county level. Panel A,** locations of SAPs detected from sandflies. **Panel B,** locations of SAPs detected from animals. **Panel C,** the average annual incidence of VL from 2014–2018 recorded at the county level. **Panel D,** the model-predicted annual incidence of VL in 2014–2018. The provinces outlined in red, green, and purple correspond to AVL&DT-ZVL endemicity, MT-ZVL endemicity, and endemicity for both types, respectively. The white area represents provinces that are non-endemic and devoid of any risk areas. SAPs sandfly-associated pathogens; VL visceral leishmaniasis; AVL&DT-ZVL anthroponotic VL and desert-type zoonotic VL; MT-ZVL mountain-type zoonotic VL. Base layers of the maps were downloaded from Resource and Environment Science and Data Center (https://www.resdc.cn/DOI/DOI.aspx?DOIID=120).

five sandfly species in 24 counties and 14 animal species in 61 counties, mainly in central and northwestern China (Fig 3A and 3B). Among the five sandfly species capable of carrying the *L. donovani* complex, four have been confirmed as competent vectors: i.e., *P. chinensis*, *P. longiductus*, *P. wui*, and *P. alexandri*. Among them, the detection of *L. donovani* complex in *P. chinensis* was reported in the highest number of counties (11 counties), followed by *P. wui* (in 10 counties), *P. alexandri* (in 6 counties) and *P. longiductus* (in 3 counties). In contrast, *P. mongolensis* is not a permissive vector for transmission of the *L. donovani* complex in China [17,23]. Dogs, as confirmed reservoirs of MT-ZVL, had been determined to carry *L. donovani* complex with the highest frequency, reported in 44 counties. Other domesticated animals other than dog, as well as wild animals, although determined to carry *L. donovani* complex (Fig 2E), had not been confirmed to be the role of reservoir host.

*L. turanica* was determined from two sandfly species in 4 counties, and from three animals in 12 counties, all restricted to northwestern China, mainly in northern Xinjiang, western

Inner Mongolia, and northwestern Gansu (Fig 3A and 3B). Two *Rickettsia* species with potential human-pathogenicity, *R. bellii* and one uncharacterized *Rickettsia* SXWX1916-2, were respectively identified from *P. chinensis* in Henan and Shanxi provinces [24,25]. Two viruses both belonging to *Phlebovirus*, WUXV and HEDV, were first determined from *P. chinensis* in Shanxi Province in 2018 (Fig 3A) [26–29]. By the end of 2022, both WUXV and HEDV viruses had been detected in multiple locations in Shanxi Province (Fig 3A and 3B).

### 3.4 Risk mapping and risk factors of VL

A total of 1,070 human cases of VL were reported from six provinces during 2014 and 2018, with most of the cases reported in western and central China, including in Xinjiang (446 cases), Gansu (349), Sichuan (104), Shanxi (82), Shaanxi (77), and Henan (12) provinces (Fig 3C). Two-stage XGBoost models were established on human cases in AVL&DT-ZVL and MT-ZVL endemic provinces respectively, which demonstrated different risk levels for the endemic counties (Table F in S1 Appendix). The first-stage models achieved an adequate performance to predict the occurrence of VL cases, with AUCs achieving 0.94 for both training and test datasets. Based on the second-stage model that predicts the incidence rate of VL, the high-risk areas during 2014–2018 were generally consistent across the distributed provinces (Xinjiang, Gansu, Shanxi, and Sichuan). However, the predicted high-risk areas were determined to be more extensive than what was observed (2,658,000 km$^2$ *vs*. 1,123,300 km$^2$) (Fig 3C and 3D and Table 2). The provinces that were predicted to be high-risk (those with at least one county with a predicted VL incidence $\geq 1.490/10^5$) by the model were consistent with the provinces that reported high case number according to the surveillance data. Two provinces, Qinghai, and Yunnan, were predicted as new endemic provinces (Fig 3C and 3D). Approximately 2,853,000 and 61,185,000 people lived in the predicted high-risk and low-medium-risk areas, which was 17.9% and 132.7% more than what was observed (2,420,000, 26,293,000) (Table 2).

For AVL&DT-ZVL, the model predicted a more extensive area with both high-risk and low-medium-risk mainly in Xinjiang. Compared with actual reports, the county number and population size in high-risk areas in Xinjiang were predicted to be increased by 75.0% and 28.9%, respectively, while those in low-medium-risk areas were predicted to be augmented by 39.3% and 56.2%, respectively (Table 2). Six significant predictors contribute to the presence of AVL&DT-ZVL. Total precipitation was the most influential predictor with a relative contribution (RC) of 23.97%. The other predictors include the presence of *P. wui* (RC of 21.07%), annual mean temperature (RC of 9.43%), standard deviation of elevation (RC of 5.83%), precipitation of the driest quarter (RC of 5.80%), and population density (RC of 5.17%). These predictors differed from those that contributed to their incidence level, which include precipitation seasonality (RC of 33.76%), standard deviation of elevation (RC of 14.15%), temperature seasonality (RC of 13.92%), annual mean temperature (RC of 12.06%) and isothermality (RC of 6.12%). Two significant predictors with RC $\geq 5\%$ overlapped between two-stage models, i.e., the annual mean temperature and standard deviation of elevation (Table 3 and Fig J in S1 Appendix).

For MT-ZVL, the XGBoost model predicted high-risk areas in provinces that largely resembled the observed regions, with significantly expanded low-medium-risk areas in Gansu, Shaanxi, Shanxi, and Sichuan provinces (Fig 3C and 3D). Compared to the observed data, the population size affected by MT-ZVL is augmented by 11.4% and 164.9% in high-risk and low-medium-risk areas, respectively (Table 2). Five shared predictors contributed significantly to both the occurrence and incidence level of MT-ZVL, including annual mean temperature, isothermality, precipitation seasonality, precipitation of driest quarter and standard deviation of elevation (Table 3 and Fig K in S1 Appendix). The average elevation (RC of 19.32%), presence

**Table 2. Projections of the counties, areas and population size potentially affected by VL cases under SSP585 in the mainland of China.**

| Province | Number of counties[#] (Relative difference %)* | | Area×10³ km²[#] (Relative difference %)* | | Population×10⁴ persons[#] (Relative difference %)* | |
|---|---|---|---|---|---|---|
| | High-risk | Low-medium-risk | High-risk | Low-medium-risk | High-risk | Low-medium-risk |
| AVL&DT-ZVL | | | | | | |
| Gansu | – | 1/2/1/2/6 (100.0/-50.0/100.0/200.0) | – | 231.9/494.6/12.6/275.4/858.6 (113.3/-97.4/2082.5/211.8) | – | 15.3/31.8/13.9/28.0/91.5 (108.2/-56.3/101.5/226.4) |
| Inner Mongolia | – | 0/0/0/0/4 (0.0/0.0/0.0/-) | – | 0.0/0.0/0.0/0.0/1628.1 (0.0/0.0/0.0/-) | – | 0.0/0.0/0.0/0.0/30.9 (0.0/0.0/0.0/-) |
| Xinjiang | 4/7/12/13/22 (75.0/71.4/8.3/69.2) | 28/39/23/25/27 (39.3/-41.0/8.7/8.0) | 822.9/2219.3/1360.9/1373.3/1921.2 (169.7/-38.7/0.9/39.9) | 5177.6/5878.3/4917.6/4887.3/5573.8 (13.5/-16.3/-0.6/14.0) | 89.9/115.9/342.1/375.6/510.5 (28.9/195.3/9.8/35.9) | 770.5/1203.7/757.6/802.3/907.1 (56.2/-37.1/5.9/13.1) |
| All | 4/7/12/13/22 (75.0/71.4/8.3/69.2) | 29/41/24/27/37 (41.4/-41.5/12.5/37.0) | 822.9/2219.3/1360.9/1373.3/1921.2 (169.7/-38.7/0.9/39.9) | 5409.5/6373.0/4930.2/5162.7/8060.6 (17.8/-22.6/4.7/56.1) | 89.9/115.9/342.1/375.6/510.5 (28.9/195.3/9.8/35.9) | 785.8/1235.5/771.5/830.3/1029.5 (57.2/-37.6/7.6/24.0) |
| MT-ZVL | | | | | | |
| Gansu | 4/4/0/0/0 (0.0/-100.0/0.0/0.0) | 16/41/0/1/4 (156.2/-100.0/-/300.0) | 155.7/155.7/0.0/0.0/0.0 (0.0/-100.0/0.0/0.0) | 517.3/1192.8/0.0/95.0/184.0 (130.6/-100.0/-/93.6) | 111.8/111.8/0.0/0.0/0.0 (0.0/-100.0/0.0/0.0) | 353.2/1271.9/0.0/44.1/100.5 (260.1/-100.0/-/127.9) |
| Hebei | – | 0/0/0/0/1 (0.0/0.0/0.0/-) | – | 0.0/0.0/0.0/0.0/26.1 (0.0/0.0/0.0/-) | – | 0.0/0.0/0.0/0.0/85.3 (0.0/0.0/0.0/-) |
| Henan | – | 3/3/0/0/0 (0.0/-100.0/0.0/0.0) | – | 19.9/19.9/0.0/0.0/0.0 (0.0/-100.0/0.0/0.0) | – | 80.7/80.7/0.0/0.0/0.0 (0.0/-100.0/0.0/0.0) |
| Qinghai | – | 0/1/0/0/0 (-/-100.0/0.0/0.0) | – | 0.0/18.6/0.0/0.0/0.0 (-/-100.0/0.0/0.0) | – | 0.0/20.4/0.0/0.0/0.0 (-/-100.0/0.0/0.0) |
| Shaanxi | – | 18/35/6/7/6 (94.4/-82.9/16.7/-14.3) | – | 341.1/758.7/136.8/148.9/139.1 (122.5/-82.0/8.8/-6.5) | – | 517.8/1079.6/146.0/174.9/90.6 (108.5/-86.5/19.8/-48.2) |
| Shanxi | 3/3/2/2/1 (0.0/-33.3/0.0/-50.0) | 14/39/4/9/16 (178.6/-89.7/125.0/77.8) | 6.5/6.5/0.7/0.7/0.4 (0.0/-89.2/0.0/-45.8) | 155.9/447.4/46.5/105.8/138.4 (187.0/-89.6/127.6/30.8) | 17.7/17.7/2.1/2.0/1.2 (0.0/-88.0/-7.6/-40.6) | 364.3/1169.3/125.8/279.4/311.6 (221.0/-89.2/122.2/11.5) |
| Sichuan | 3/5/2/1/1 (66.7/-60.0/-50.0/0.0) | 20/32/0/0/0 (60.0/-100.0/0.0/0.0) | 138.3/276.5/97.3/54.4/54.4 (100.0/-64.8/-44.1/0.0) | 648.9/723.9/0.0/0.0/0.0 (11.6/-100.0/0.0/0.0) | 22.7/40.0/16.4/9.0/6.7 (76.5/-58.9/-45.5/-25.7) | 527.5/1230.4/0.0/0.0/0.0 (133.2/-100.0/0.0/0.0) |
| Yunnan | – | 0/2/0/0/0 (-/-100.0/0.0/0.0) | – | 0.0/187.3/0.0/0.0/0.0 (-/-100.0/0.0/0.0) | – | 0.0/30.8/0.0/0.0/0.0 (-/-100.0/0.0/0.0) |
| All | 10/12/4/3/2 (20.0/-66.7/-25.0/-33.3) | 71/153/10/17/27 (115.5/-93.5/70.0/58.8) | 300.4/438.6/98.0/55.1/54.8 (46.0/-77.7/-43.8/-0.6) | 1683.1/3348.7/183.3/349.7/487.6 (99.0/-94.5/90.7/39.4) | 152.1/169.5/18.6/10.9/7.8 (11.4/-89.1/-41.1/-28.4) | 1843.5/4883.0/271.7/498.4/588.0 (164.9/-94.4/83.4/18.0) |
| Total | 14/19/16/16/24 (35.7/-15.8/0.0/50.0) | 100/194/34/44/64 (94.0/-82.5/29.4/45.5) | 1123.3/2658.0/1458.9/1428.4/1976.0 (136.6/-45.1/-2.1/38.3) | 7092.6/9721.6/5113.5/5512.4/8548.1 (37.1/-47.4/7.8/55.1) | 242.0/285.3/360.6/386.6/518.3 (17.9/26.4/7.2/34.1) | 2629.3/6118.5/1043.2/1328.7/1617.5 (132.7/-82.9/27.4/21.7) |

[#]The number, land area, and population size of counties affected by VL that were actually reported during 2014–2018, projected during 2014–2018, 2021–2040, 2041–2060, and 2061–2080, with values for different time periods separated by slashes.

*The relative differences (%) was estimated for the predicted values as compared to those of the previous period, i.e., from actual observations to projection during 2014–2018, from the projection during 2014–2018 to 2021–2040, from the projection during 2021–2040 to 2041–2060, and from the projection during 2041–2060 to 2061–2080, that were intermitted by slash in the parentheses.

"–" indicates that the predicted values for the counties, areas, or population size affected by VL cases are zero for all periods.

VL visceral leishmaniasis; SSP Shared Socioeconomic Pathway; AVL&DT-ZVL anthroponotic VL and desert-type zoonotic VL; MT-ZVL mountain-type zoonotic VL.

**Table 3. The relative contributions of major factors to the spatial distributions of two types of VL, estimated by two-stage XGBoost models.**

| Category | Variable | Relative contributions (%) mean (standard deviation) | | | |
|---|---|---|---|---|---|
| | | AVL&DT-ZVL | | MT-ZVL | |
| | | Stage 1 | Stage 2 | Stage 1 | Stage 2 |
| Socioeconomic | Rural population proportion | 4.59 (1.13) | 4.43 (2.56) | 8.25 (0.85) | 3.16 (0.28) |
| | Population density | 5.17 (1.09) | 4.84 (1.38) | 5.50 (1.79) | 3.24 (0.91) |
| | Rural population density | 3.85 (0.92) | 1.69 (1.47) | 4.67 (0.63) | 4.98 (0.59) |
| Bioclimatic | Annual mean temperature | 9.43 (1.91) | 12.06 (5.12) | 6.31 (0.91) | 21.76 (3.96) |
| | Mean diurnal range | 4.54 (1.03) | | 6.45 (0.78) | 3.97 (0.51) |
| | Isothermality | 4.12 (1.05) | 6.12 (1.13) | 6.52 (1.33) | 6.41 (3.56) |
| | Temperature seasonality | 3.53 (0.79) | 13.92 (4.38) | 7.67 (0.98) | 4.10 (1.10) |
| | Max temperature of warmest month | 3.44 (0.84) | | 4.58 (0.91) | 4.88 (2.29) |
| | Total precipitation | 23.97 (5.73) | 3.26 (0.50) | 4.94 (0.70) | 8.40 (2.71) |
| | Precipitation seasonality | 2.89 (0.90) | 33.76 (10.24) | 5.69 (0.81) | 7.79 (2.92) |
| | Precipitation of driest quarter | 5.80 (1.21) | 4.88 (2.00) | 6.56 (0.87) | 9.65 (4.71) |
| Terrain | Standard deviation of elevation | 5.83 (0.89) | 14.15 (3.76) | 5.64 (1.14) | 17.61 (2.01) |
| | Average elevation | 1.75 (0.63) | 0.88 (0.47) | 19.32 (2.54) | 4.07 (1.29) |
| Sandfly vector | Presence of *P. chinensis* | | | 8.36 (1.75) | |
| | Presence of *P. wui* | 21.07 (4.49) | | | |

Stage 1 models the presence/absence of reported human case, and stage 2 models the annual average incidences from 2014 to 2018 within the presence locations. VL visceral leishmaniasis; XGBoost extreme gradient boosting; AVL&DT-ZVL anthroponotic VL and desert-type zoonotic VL; MT-ZVL mountain-type zoonotic VL.

of *P. chinensis* (RC of 8.36%), rural population proportion (RC of 8.25%), temperature seasonality (RC of 7.67%) each contributed exclusively to the presence of MT-ZVL. Compared to the first-stage model, annual mean temperature (RC of 21.76%) and standard deviation of elevation (RC of 17.61%) contributed more significantly to the incidence of MT-ZVL in the model (Table 3).

## 3.5 Mapping future risk and changes of VL incidence under the SSPs

We predicted the average incidences of AVL&DT-ZVL and MT-ZVL under three scenarios across four time periods: 2014–2018, 2021–2040, 2041–2060, and 2061–2080. Overall, we predicted central and western China remaining as VL risk areas in the future, however, showing differential changing trends (Fig 4 and Figs L and M in S1 Appendix). For example, under the SSP585 scenario, the number, area, and population of high-risk counties in northwestern China affected by AVL&DT-ZVL are projected to gradually increase in each future period. Conversely, those in central China affected by MT-ZVL are projected to gradually decrease in each future period (Fig 4 and Table 2).

For the human cases with AVL&DT-ZVL, we predicted only minor differences in the risk areas under the SSP126 and SSP245 scenarios (Figs L and M in S1 Appendix). In both scenarios, the number of counties at high-risk of disease exhibited an increased trend during the 2021–2040, with ensuing stabilization in the future, while the number of counties at low-medium-risk was predicted to be decreased during the 2021–2040 period and then stabilize at comparable level (Figs L and M in S1 Appendix, Tables I and G in S1 Appendix). By contrast, under the SSP585 scenario, the number of high-risk counties was predicted to increase persistently, with 1,312,000 and 587,000 people additionally to be affected in high-risk areas during the 2061–2080 period, when compared to SSP126 and SSP245 scenarios, respectively. The low-medium-risk area was predicted to be established in more extensive ranges under the

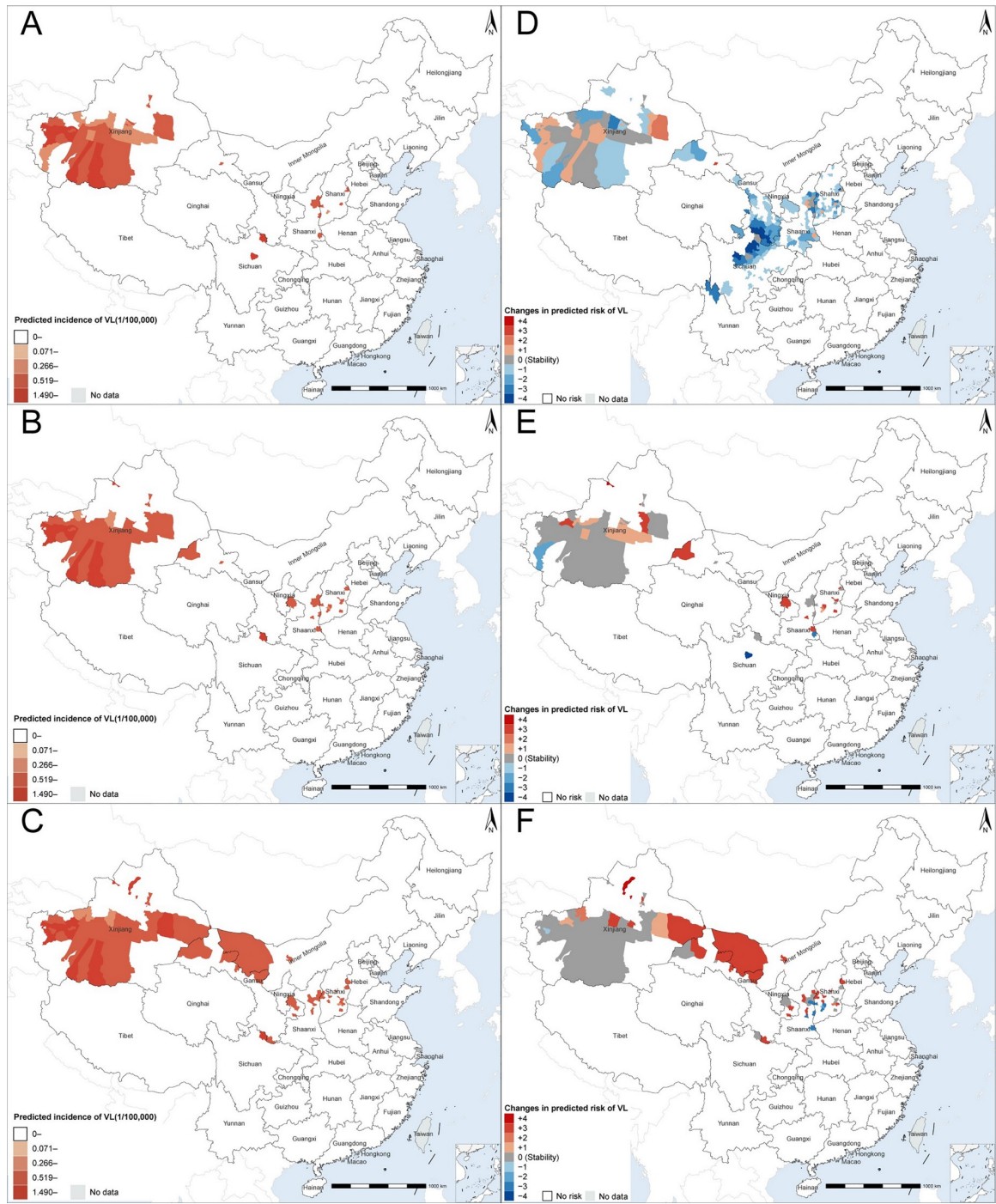

**Fig 4. Spatial distribution and changes in model-predicted incidence of VL under SSP585. Panels A–C,** predicted annual incidence of VL in 2021–2040, 2041–2060, and 2061–2080, respectively. **Panels D–F,** changes in predicted VL risk level from 2014–2018 to 2021–2040, from 2021–2040 to 2041–2060, and from 2041–2060 to 2061–2080, respectively. The white block labeled "no risk" indicates that the VL risk level has not changed compared to the previous period, and both periods are predicted to be "no risk". VL risk levels were divided into five levels based on quartiles of county-level annual incidences from 2014 to 2018: level 0 (no risk), level 1 (>0 but <25th percentile), level 2 (≥25th percentile but <50th percentile), level 3 (≥50th percentile but <75th percentile), and level 4 (≥75th percentile). The change in VL risk level was calculated as the predicted VL risk level of a specific county in a later period minus the predicted VL risk level in the previous period. VL visceral leishmaniasis. Base layers of the maps were downloaded from Resource and Environment Science and Data Center (https://www.resdc.cn/DOI/DOI.aspx?DOIID=120).

SSP585 scenario than under the SSP126 and SSP245 (Table 2). In all three future scenarios, the high-risk areas were predicted to aggregate in Xinjiang, showing an increasing trend. The population affected in these high-risk areas was projected to reach 3,793,000 (under SSP126), 4,518,000 (under SSP245), and 5,105,000 (under SSP585), respectively, by 2080 (Table 2 and Tables I and J in S1 Appendix).

For the human cases with MT-ZVL, the predicted areas at risk demonstrated similar changes under three scenarios, all showing a substantial decrease in the number of at-risk counties during the 2021–2040 period. There was a continuous decrease in the number of high-risk counties, while concurrently, a slight increase in the number of low-medium-risk counties was observed from the period of 2021–2040 to 2061–2080 (Fig 4 and Figs L and M in S1 Appendix). Taking the SSP585 scenario as an example, the high-risk areas were predicted to be present in Sichuan, Shanxi, and Gansu provinces for all three periods spanning from 2014–2018 to 2061–2080. Furthermore, the affected population in the high-risk areas might experience a decay by over 28% in each period compared to the previous one (i.e., 2021–2040 relative to 2014–2018, 2041–2060 relative to 2021–2040, and 2061–2080 relative to 2041–2060). The models predict a significantly decreased population size at low-medium-risk from 48,830,000 in 2014–2018, to 2,717,000 in 2021–2040, before its gradual increase to 4,984,000 in 2041–2060 and 5,880,000 in 2061–2080 (Table 2).

## 4. Discussion

We have assembled the most comprehensive records of sandflies and SAPs in the mainland of China, mapping their locations at the county level over the past 80 years. We established a robust machine learning algorithm and predict much broader areas that are suitable habitat for sandfly species, particularly for the four sandfly vectors for VL (all showing relative differences above 41%) (Table 1). This difference observed between the predictions and the actual observation is largely owing to the limited field surveys and incomplete sampling of sandfly and SAPs in China. The overestimation of the current models might also be responsible.

Given the wide distribution and complex ecological niches of sandflies, and the heterogeneous predictors shown within the same genus, we further group the major sandfly species into three clusters. Within each cluster, similar ecological niches and geographic distributions are presented, which may help to pinpoint potential risk areas for disease transmission and guide health authorities in determining where to focus surveillance efforts and where to use preventive measures. For example, if a target sandfly species has a low identification rate in a specific area, but other species within the same cluster are proved to be prevalent in that area, then surveying of the target sandfly species and the SAPs related to this sandfly species should also be prioritized in that area.

Of all the predictors, total precipitation, temperature seasonality, and annual temperature contribute with the greatest effect to the distribution of sandfly species, highly likely through affecting their suitable breeding habitat. For instance, the predicted distribution of sandflies in Cluster 2 is almost entirely located in northwestern China, which is featured by low precipitation and extensive desert areas. The caves in the desert, which have moderate temperature/humidity, inhabited by small rodents and reptiles, could provide both shelter and shade and blood source for the survival and development of sandflies [14]. Experimental studies have fully confirmed the dramatic impact of temperature on all stages of sandfly development. For instance, in a suitable environment with sufficient food available, larval development of sandfly is generally completed within 20–30 days, depending on the species [30]. However, this period may be prolonged to several months if the sandfly enter diapause to cope with low-temperature conditions [14].

Globally, in addition to the *L. donovani* complex, sandflies also carry multiple viruses within the *Phlebovirus* genus. These viruses typically cause uncomplicated to moderate fever episodes, commonly referred to as "sandfly fever" or "pappataci fever" when identified [7,8]. Other SAPs with human pathogenic property, including bacteria from the genera *Bartonella* and *Coxiellaceae* [24,31], and viruses from the genera *Vesiculovirus* and *Orbivirus* are present as well [32]. However, in China, investigations of SAPs are almost solely on *Leishmania*, with only recent high-throughput sequencing efforts made to discover viruses and bacteria [26–29]. Remarkably, two distinct virus species, WUXV and HEDV, closely related to the human-pathogenic members in the genus *Phlebovirus*, were determined. WUXV was first isolated from *P. chinensis* in Wuxiang County, Shanxi Province, in 2018 [27]. Phylogenetic analysis shows that WUXV is most closely related to Toros virus (TORV) and Corfou virus (CFUV), causing endemic diseases in the Mediterranean region [27]. Although no human patient with WUXV infection has been reported, the specific antibodies were determined in healthy residents and chickens [33]. HEDV, a close relative of Rift Valley fever virus (RVFV), was first determined through meta-transcriptomics sequencing rom *P. chinensis* in 2018 in Yangquan City, Shanxi Province, which was subsequently detected in multiple locations in the same Province. Both WUXV and HEDV can be isolated and grown in mammalian cell lines [26,28], indicating the potential of these novel sandfly-associated viruses in causing human infection.

Our two-stage modelling effort has disclosed different factors that contributed to the presence and incidence level of human cases with VL, however with the annual mean temperature and standard deviation of elevation overlapping between two stages. The current two-stage modelling represents a more precise predictive efforts, particularly when the vector presence was incorporated as a predictor in the ecological model, allowing us to assess the net effect of other factors on VL while controlling for their impact on the ecology of eligible vectors.

Our results also indicate that the DT-ZVL transmitted by *P. wui* and the MT-ZVL transmitted by *P. chinensis* are currently the dominant VL types, in agreement with several previous studies [10,17]. Separate modeling revealed accurate predictions for the two disease types, which also mimic the predicted distribution of their corresponding vectors, i.e., *P. chinensis* and *P. wui* (Fig 3C and 3D and Fig G in S1 Appendix). Active vector surveillance may serve as an effective measure for alerting to the human VL cases.

Our study provides a framework for assessing the impact of climate change and other non-biological factors on the spread of VL disease. This framework could be applied to other regions and scenarios, thereby enhancing our understanding and prediction of future trends in VL. For instance, our findings highlight that under the SSP585 scenario, due to higher emissions and a fossil-fuel intensive pathway, the expansion risk of VL is greatest. Conversely, under the SSP126 and SSP245 scenarios, which are characterized by lower emissions, the expansion risk of VL is reduced. These findings underscore the importance of considering the impacts of climate change under different scenarios when formulating strategies for VL prevention and control. The effect of climate changes on the VL disease is also estimated in the wider context of a set of non-biological factors, such as terrain and socioeconomic development. We demonstrate a significant decrease of MT-ZVL incidence, in contrast to an expansion of AVL&DT-ZVL in future. Moreover, the expansion risk is greatest in the SSP585 scenario compared to the SSP126 and SSP245 scenarios. These results indicate differential impacts from climate change that can exert on the different types VL in China, depending on the regional characteristics and the future emission scenarios. However, the VL trend in central China remains relatively stable across different scenarios, which may indicate that other non-biological factors, such as terrain and socio-economic development, also modulate the effect of climate change on VL. Therefore, the projection and control efforts of VL necessitate a full consideration of both the biological and non-biological factors, which should also be

tailored to different regions and scenarios. Policymakers and public health officials can leverage this information to prioritize areas at greater risk and allocate resources more effectively.

There are several limitations to our study. First, the locations of the sandfly surveys were extracted from the literature, which were unlikely to be randomly sampled. Although the surveyed counties were highly representative (Fig A in S1 Appendix), and the probability of being surveyed counties were weighted, the potential bias remained in the BRT model. Second, both the BRT and XGBoost models are prone to overfitting [21,34]. The solution to overfitting is to optimally adjust the model configuration parameters through cross-validation [35]. However, we could not perform a complete cross-validation optimization of all model configuration parameters due to the large scale and number of model runs.

## 5. Conclusions

Our study underscores the necessity for a comprehensive and forward-thinking approach in managing sandflies and VL management, considering the intricate ecology and potential ramifications of global changes. We quantify the effects of different risk factors on VL incidence and sandfly distribution, highlighting the significant influence of different vectors on the incidence of various types of VL. Our study highlights the currently underestimated geographic range and disease burden of sandfly and SAPs in China. Furthermore, our study projects the potential risk and geographic spread of VL under future global changes, with a particular focus on northwestern China, where the risk of VL is high and anticipated to persist. We recommend enhanced sandfly surveillance efforts, improved VL diagnostic technology, and elevated public awareness. Targeted prevention and control measures should be implemented based on regional and future scenario predictions to effectively mitigate the impact of VL.

## Supporting information

**S1 Appendix. Text A.** Data Collection and Management. **Text B.** Ecological modeling and clustering analysis of sandflies**. Text C.** Ecological Modeling of VL. **Fig A.** The spatial distribution of the 673 counties with at least one record of sandflies (yellow) from 1940 to 2022, China. Base layers of the maps were downloaded from Resource and Environment Science and Data Center (https://www.resdc.cn/DOI/DOI.aspx?DOIID=120). **Fig B.** The number of records related to sandflies in different provinces in different periods (with the number of publications in parentheses) and a comparison of the total number of records for four VL vector sandflies in each province. **Fig C.** The spatial distribution of the sandfly genus Phlebotomus recorded at the county level from 1940 to 2022 in China. Base layers of the maps were downloaded from Resource and Environment Science and Data Center (https://www.resdc.cn/DOI/DOI.aspx?DOIID=120). **Fig D.** The spatial distribution of the sandfly genus *Sergentomyia* recorded at the county level from 1940 to 2022 in China. Base layers of the maps were downloaded from Resource and Environment Science and Data Center (https://www.resdc.cn/DOI/DOI.aspx?DOIID=120). **Fig E.** The spatial distribution of the sandfly genus *Chinius* and *Idiophlebotomus* recorded at the county level from 1940 to 2022 in China. Base layers of the maps were downloaded from Resource and Environment Science and Data Center (https://www.resdc.cn/DOI/DOI.aspx?DOIID=120). **Fig F.** Sandfly species richness (circles) at the prefecture level in seven biogeographic zones in the mainland of China from 1940 to 2022. Base layers of the maps were downloaded from Resource and Environment Science and Data Center (https://www.resdc.cn/DOI/DOI.aspx?DOIID=120). **Fig G.** The predicted county-level distributions of the six most prevalent sandfly species in the *Phlebotomus* genus, averaged over the ensemble of BRT models. Base layers of the maps were downloaded from Resource and Environment Science and Data Center (https://www.resdc.cn/DOI/DOI.aspx?DOIID=120). **Fig H.** The predicted county-level

distributions of the six most prevalent sandfly species in the *Sergentomyia* genus, averaged over the ensemble of BRT models. Base layers of the maps were downloaded from Resource and Environment Science and Data Center (https://www.resdc.cn/DOI/DOI.aspx?DOIID=120). **Fig I.** The mean curves (red) and 95% percentiles (gray) for the effects of major predictors (RC ≥5%) on the probability of occurrence of 12 main sandfly species based on the ensemble of BRT models. **Fig J.** XGBoost-model-predicted AVL and DT-ZVL incidence in response to major predictors (RC ≥5%) when other predictors are fixed at mean values. **Fig K.** XGBoost-model-predicted MT-ZVL incidence in response to major predictors (RC ≥5%) when other predictors are fixed at mean values. **Fig L.** Spatial distribution and changes in model-predicted incidence of VL in the mainland of China under SSP126. Base layers of the maps were downloaded from Resource and Environment Science and Data Center (https://www.resdc.cn/DOI/DOI.aspx?DOIID=120). **Fig M.** Spatial distribution and changes in model-predicted incidence of VL in the mainland of China under SSP245. Base layers of the maps were downloaded from Resource and Environment Science and Data Center (https://www.resdc.cn/DOI/DOI.aspx?DOIID=120). **Fig N.** Spatial distribution and changes in model-predicted environmental suitability of *P. wui* in future under three scenarios. Base layers of the maps were downloaded from Resource and Environment Science and Data Center (https://www.resdc.cn/DOI/DOI.aspx?DOIID=120). **Fig O.** Spatial distribution and changes in model-predicted environmental suitability of *P. chinensis* in future under three scenarios. Base layers of the maps were downloaded from Resource and Environment Science and Data Center (https://www.resdc.cn/DOI/DOI.aspx?DOIID=120). **Fig P.** Spatial distribution and changes in model-predicted environmental suitability of *P. longiductus* in future under three scenarios. Base layers of the maps were downloaded from Resource and Environment Science and Data Center (https://www.resdc.cn/DOI/DOI.aspx?DOIID=120). **Fig Q.** Spatial distribution and changes in model-predicted environmental suitability of *P. alexandri* in future under three scenarios. Base layers of the maps were downloaded from Resource and Environment Science and Data Center (https://www.resdc.cn/DOI/DOI.aspx?DOIID=120). **Table A.** The specific references for all 47 sandfly species in China from 1940 to 2022. **Table B.** The inclusion and exclusion criteria for screening publications. **Table C.** Original resolutions and extents of source datasets. **Table D.** Potential risk factors at the county level used in the BRT model for sandfly species and 2-stage XGBoost model for VL. **Table E.** Clustering analysis of model predictors at the county level based on pairwise Pearson correlation coefficients. **Table F.** Cross-tabulation of observed and XGBoost-model-predicted annual incidence levels of VL in 2016. **Table G.** BRT-model-estimated mean (standard deviation) relative contributions of top factors (RC ≥5%) to the spatial distribution of six most prevalent sandfly species in the *Phlebotomus* genus. **Table H.** BRT-model-estimated mean (standard deviation) relative contributions of top factors (RC ≥5%) to the spatial distribution of six most prevalent sandfly species in the *Sergentomyia* genus. **Table I.** Projections of the numbers, land areas, and population sizes of counties affected by VL risk areas according to SSP126. **Table J.** Projections of the numbers, land areas, and population sizes of counties affected by VL risk areas according to SSP245.
(PDF)

**S1 Data. The distribution of sandflies recoreded at the county levels.**
(XLSX)

## Acknowledgments

The authors thank the personnel who investigated and reported the distribution of sandflies, as well as the Chinese Center for Disease Control and Prevention for providing the VL surveillance data.

## Author Contributions

**Conceptualization:** Xue-Geng Hong, Yi Sun, Li-Qun Fang, Wei Liu.

**Data curation:** Xue-Geng Hong, Ying Zhu, Tao Wang, Jin-Jin Chen, Fang Tang, Rui-Ruo Jiang, Xiao-Fang Ma, Qiang Xu, Hao Li, Li-Ping Wang.

**Formal analysis:** Xue-Geng Hong, Ying Zhu.

**Funding acquisition:** Li-Qun Fang, Wei Liu.

**Methodology:** Xue-Geng Hong, Ying Zhu, Tao Wang, Jin-Jin Chen, Hao Li, Yi Sun, Li-Qun Fang, Wei Liu.

**Project administration:** Yi Sun, Li-Qun Fang, Wei Liu.

**Resources:** Li-Ping Wang.

**Software:** Xue-Geng Hong, Ying Zhu, Tao Wang.

**Supervision:** Yi Sun, Li-Qun Fang, Wei Liu.

**Validation:** Qiang Xu.

**Visualization:** Xue-Geng Hong, Ying Zhu, Fang Tang, Rui-Ruo Jiang, Xiao-Fang Ma.

**Writing – original draft:** Xue-Geng Hong, Ying Zhu.

**Writing – review & editing:** Li-Qun Fang, Wei Liu.

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
