## [Decision Letter · Decision Letter 0]

30 May 2024

Dear Dr. Liu,

Thank you very much for submitting your manuscript "Mapping the distribution of sandflies and sandfly-associated pathogens in China" for consideration at PLOS Neglected Tropical Diseases. As with all papers reviewed by the journal, your manuscript was reviewed by members of the editorial board and by several independent reviewers. The reviewers appreciated the attention to an important topic. Based on the reviews, we are likely to accept this manuscript for publication, providing that you modify the manuscript according to the review recommendations. 

Sincerely,

Luc E. Coffeng, MD PhD

Academic Editor

Amy Morrison

Section Editor

Reviewer's Responses to Questions

**Key Review Criteria Required for Acceptance?**

**Methods**

-Are the objectives of the study clearly articulated with a clear testable hypothesis stated?

-Is the study design appropriate to address the stated objectives?

-Is the population clearly described and appropriate for the hypothesis being tested?

-Is the sample size sufficient to ensure adequate power to address the hypothesis being tested?

-Were correct statistical analysis used to support conclusions?

-Are there concerns about ethical or regulatory requirements being met?

Reviewer #1: (No Response)

Reviewer #2: The objectives and study design look fine. The clarity, appropriateness and size of the population under study are based largely on those from previously accepted publications. Sophisticated and advanced statistical programs were used for data analysis to support the conclusions. There are no ethical or regulatory concerns.

**Results**

-Does the analysis presented match the analysis plan?

-Are the results clearly and completely presented?

-Are the figures (Tables, Images) of sufficient quality for clarity?

Reviewer #1: (No Response)

Reviewer #2: There are ample and detailed descriptions presented according to analysis plan. For the most part, results are amply presented in some detail supported by figures with reasonable clarity.

**Conclusions**

-Are the conclusions supported by the data presented?

-Are the limitations of analysis clearly described?

-Do the authors discuss how these data can be helpful to advance our understanding of the topic under study?

-Is public health relevance addressed?

Reviewer #1: (No Response)

Reviewer #2: The data presented are in support of conclusions for the most part Some clear limitations of the analysis were listed and described. The potential applications of the approaches were stated with relevance to public health.

**Editorial and Data Presentation Modifications?**

Reviewer #1: (No Response)

Reviewer #2: The Abstract needs attention to make sure that it covers all the essential points of the work presented. For example, 6 known SAPs (sand fly-associated pathogens) were mentioned in the text, but only Leishmania appeared in the abstract.

**Summary and General Comments**

Reviewer #1: This study comprehensively investigates the distribution of sandflies and sandfly-associated pathogens (SAPs) in China, as well as the incidence risk of human visceral leishmaniasis (VL). By analyzing literature data from 1940 to 2022 and surveillance data from 2014 to 2018, the research maps out the distribution of sandfly species and identifies ecological drivers using machine learning techniques. The findings indicate a significant underestimation of current sandfly distribution and VL risk, and predicted their future distribution and risk under three scenarios of climate and socioeconomic changes. The study highlights the need for enhanced surveillance and field investigations, particularly in regions where VL risk is projected to persist or increase amidst changing climate and socioeconomic scenarios. Overall, I think the paper offers valuable insights into an important area of neglected tropical diseases. I have several general comments:

1. The first paragraph of the Introduction is too general; please reshape the text to improve readability. Additionally, in the second paragraph of the Introduction, please introduce the situation of sandfly-associated pathogens and visceral leishmaniasis in China.

2. In both Fig. 3 a-c, the provinces of Qinghai and Yunnan are not mentioned. I don’t quite understand why the VL annual incidence predicted by the model from 2014 to 2018 would lead to Qinghai and Yunnan becoming new endemic provinces. Please provide more detailed interpretation of this result. Furthermore, please clarify the meaning of the white color in the legend of Fig 3-4. For instance, in Fig 4 a-d, the white blocks are described as “0” or “no risk”. Is this because the reported data is actually zero, or because there is no data available? If the reported data for these provinces is zero, please clarify. If it is because there is no data available, the description is incorrect. 

3. This is a good piece of work, but I believe the discussion section does not fully illustrate on the significance of this study. For example, in lines 525-526, the authors wrote, “Moreover, the expansion risk is greatest in the SSP585 scenario compared to the SSP126 and SSP245 scenarios.” This is not surprising because SSP585 inherently represents a high-emission scenario with a fossil-fuel intensive pathway. Currently, there is not enough discussion on the differences in results under different scenarios and the significance of conducting this analysis. I believe the authors should reorganize the discussion section to highlight the universal value of this study.

Specific Comments

• Some of the keywords are too general; please use more appropriate terms.

• Lines 125-126: Please clearly state what the final selection of 8 bioclimatic variables is.

• Lines 340: Should it be six provinces instead of seven provinces? 

• Lines 408-410: The description of the trend in VL incidence is not accurate. Due to the inconsistent trends in VL incidence under different Shared Socioeconomic Pathways scenarios and time periods, please clarify which specific scenario and time period the changes are based on.

• Lines 530-531: The description here is repetitive compared to lines 527-528.

• Please reorganize the conclusion section. It seems too vague and does not highlight the key points of this study.

• Please carefully review the format of the references, some of which is not consistent.

Reviewer #2: (No Response)

PLOS authors have the option to publish the peer review history of their article (what does this mean?). If published, this will include your full peer review and any attached files.

Reviewer #1: No

Reviewer #2: No

Figure Files:

Data Requirements:

Reproducibility:

References

---

## [Decision Letter · Decision Letter 1]

13 Jun 2024

Dear Dr. Liu,

We are pleased to inform you that your manuscript 'Mapping the distribution of sandflies and sandfly-associated pathogens in China' has been provisionally accepted for publication in PLOS Neglected Tropical Diseases.

Best regards,

Luc E. Coffeng, MD PhD

Academic Editor

Amy Morrison

Section Editor

Reviewer's Responses to Questions

**Key Review Criteria Required for Acceptance?**

**Methods**

-Are the objectives of the study clearly articulated with a clear testable hypothesis stated?

-Is the study design appropriate to address the stated objectives?

-Is the population clearly described and appropriate for the hypothesis being tested?

-Is the sample size sufficient to ensure adequate power to address the hypothesis being tested?

-Were correct statistical analysis used to support conclusions?

-Are there concerns about ethical or regulatory requirements being met?

Reviewer #1: (No Response)

**Results**

-Does the analysis presented match the analysis plan?

-Are the results clearly and completely presented?

-Are the figures (Tables, Images) of sufficient quality for clarity?

Reviewer #1: (No Response)

**Conclusions**

-Are the conclusions supported by the data presented?

-Are the limitations of analysis clearly described?

-Do the authors discuss how these data can be helpful to advance our understanding of the topic under study?

-Is public health relevance addressed?

Reviewer #1: (No Response)

**Editorial and Data Presentation Modifications?**

Reviewer #1: (No Response)

**Summary and General Comments**

Reviewer #1: The author has well addressed my concerns.

PLOS authors have the option to publish the peer review history of their article (what does this mean?). If published, this will include your full peer review and any attached files.

Reviewer #1: No

---

## [Editor Report · Acceptance letter]

1 Jul 2024

Dear Dr. Liu,

We are delighted to inform you that your manuscript, "Mapping the distribution of sandflies and sandfly-associated pathogens in China," has been formally accepted for publication in PLOS Neglected Tropical Diseases.

Best regards,

Shaden Kamhawi

co-Editor-in-Chief

Paul Brindley

co-Editor-in-Chief
